# Deep Learning of Phase-Contrast Images of Cancer Stem Cells Using a Selected Dataset of High Accuracy Value Using Conditional Generative Adversarial Networks

**DOI:** 10.3390/ijms24065323

**Published:** 2023-03-10

**Authors:** Zaijun Zhang, Hiroaki Ishihata, Ryuto Maruyama, Tomonari Kasai, Hiroyuki Kameda, Tomoyasu Sugiyama

**Affiliations:** 1School of Bioscience and Biotechnology, Tokyo University of Technology, 1401-1 Katakura-Machi, Hachioji, Tokyo 192-0982, Japan; 2School of Computer Science, Tokyo University of Technology, 1401-1 Katakura-Machi, Hachioji, Tokyo 192-0982, Japan; 3Neutron Therapy Research Center, Okayama University, 2-5-1 Shikada-cho, Kita-ku, Okayama 700-8558, Japan

**Keywords:** artificial intelligence, cancer stem cell, cell morphology, segmentation, CNN, CGAN

## Abstract

Artificial intelligence (AI) technology for image recognition has the potential to identify cancer stem cells (CSCs) in cultures and tissues. CSCs play an important role in the development and relapse of tumors. Although the characteristics of CSCs have been extensively studied, their morphological features remain elusive. The attempt to obtain an AI model identifying CSCs in culture showed the importance of images from spatially and temporally grown cultures of CSCs for deep learning to improve accuracy, but was insufficient. This study aimed to identify a process that is significantly efficient in increasing the accuracy values of the AI model output for predicting CSCs from phase-contrast images. An AI model of conditional generative adversarial network (CGAN) image translation for CSC identification predicted CSCs with various accuracy levels, and convolutional neural network classification of CSC phase-contrast images showed variation in the images. The accuracy of the AI model of CGAN image translation was increased by the AI model built by deep learning of selected CSC images with high accuracy previously calculated by another AI model. The workflow of building an AI model based on CGAN image translation could be useful for the AI prediction of CSCs.

## 1. Introduction

Cancer Stem Cells (SCs; CSCs) are a minor group of cancer cells that induce new tumors and play an important role in cancer progression. Although the nature of CSCs remains unclear, it has become increasingly obvious that CSCs have characteristics of plasticity, metabolism, and immune dynamics, and resultant resistance to cancer treatments [1,2,3,4]. CSCs are produced through cell conversion from non-CSCs due to environmental stimuli. This concept is updated from the original CSC concept and is supported by xenotransplantation models. Mitochondrial morphological dynamism is associated with CSC stemness and cell viability. CSCs possess the molecular mechanism of evading tumor immune surveillance. Therapeutic strategies have been proposed to target CSCs and their niches.

The discovery of unique surface markers on CSCs in blood tumors has accelerated CSC research on solid tumors [1]. Most tumors harbor cells with markers whose expression levels correlate with overall survival [5]. Thus, it is feasible to examine the presence of CSCs for tumor diagnosis and therapeutic development. In this context, CSC morphology must be examined extensively. It is well known that embryonic stem (ES) cells and induced pluripotent stem (iPS) cells show distinctively small cell sizes, a small cytoplasmic/nuclear ratio, clear colony borders, and a piled colony compared to non-ES and non-iPS cells [6]. In a CSC model, cells converted from iPS cells under conditioned culture medium of cancer cells are morphologically similar to iPS cells [7]. SC morphology is accompanied by cell proliferation and signaling in bone-marrow-derived mesenchymal SCs [8]. Awareness of these features largely depends on the skills of SC and CSC culture experts.

Over the past few years, artificial intelligence (AI) technology for image recognition has achieved remarkable advances. AI technologies have been applied in cell biology for the automation of intricate biological image recognition, which is only possible for skilled trainees with expertise. For example, in recent studies using microscope cell images, an AI model distinguished keratinocyte SC colonies from non-SC colonies [9]. AI accurately classified the state of cultured iPS cells into colonies, single cells, differentiated cells, and dead cells, as performed by a trainee with iPS cell culture expertise [10]. An AI model predicted the differentiation status of nephrogenic progenitor cells in kidney organoids derived from iPS cells [11]. These AI models demonstrate the capability of AI to recognize SC morphology. Using flow cytometer-derived single-cell images, whose morphology had not been examined by SC experts, an AI model identified differentiated neurons derived from neural SCs based on their morphology [12]. Expertise in SC morphology in conventional microscopy is valuable for the preparation of training datasets. This study showed that such knowledge might not be required if the presence of morphological differences between SCs and non-SCs is established.

Deep learning workflows typically require annotated images for training datasets by SC and CSC experts. Here, image-to-image translation was applied with the conditional generative adversarial network (CGAN) [13], an advanced AI system that does not require settings of deep learning and image datasets to be inspected by experts for the construction of CSC recognition for the first time. In this study, the potential of AI models to detect CSCs in phase-contrast images was demonstrated. AI models recognized different culture conditions of CSCs, although both CSC types shared common stemness, rendering the collection of training datasets for an improved AI model [14]. The visualization technology of AI processing suggests the involvement of the nucleus for recognition. Herein, the examination of training datasets was extended, and significant accuracy of CSC depiction from phase-contrast cell images by AI models was shown compared to that of previous models.

## 2. Results

### 2.1. Effect of Nucleus Images on the Deep Learning of CSC Images for Image-to-Image Translation

The authors of the present study previously showed that phase-contrast tumor tissue images overlaid with nucleus images are effective as training datasets for increasing the quality of AI models compared to those without [15]. Next, the question of whether this method is applicable to images of cultured CSCs was examined. The stemness of CSCs was investigated using green fluorescent protein (GFP) from *Nanog-GFP* reporter gene-harboring mouse-induced pluripotent stem (miPS)-Lewis lung cancer (LLC)cm cells. In a previous study conducted by the authors, cells were cultured in 96-well plates [14]; however, in the present study, cells were cultured in 6 cm dishes to avoid the background gradient image from the center to the periphery in a 96-well plate. Fluorescence images of Hoechst 33342 and GFP as well as phase contrast were acquired at 100 randomly selected positions on the dish. The sets of images were processed to obtain 1000 pairs of 224 × 224 px for AI training using pix2pix, and 1000 pairs for the evaluation of the trained AI model. The AI outputs (Figure 1a, upper and lower middle), which were generated from the input image of phase contrast (Figure 1a, upper left) and of phase contrast overlaid with Hoechst 33342 (Figure 1a, lower left), were compared. The output images were similar but not identical. For example, a fluorescent cell in the target (Figure 1a, arrow) was depicted in the output image from phase-contrast input but not from phase contrast overlaid with Hoechst 33342. It was vice versa for an another fluorescent cell in the target (Figure 1a, arrow). The similarity values of recall (Figure 1b), precision (Figure 1c), and F-measure (Figure 1d) were then calculated between the output and the target against the input. Phase-contrast CSC images overlaid with Hoechst 33,42 for training AI did not show a significant increase in the F-measure value (Figure 1d). The similarity values between the training datasets were comparable.

### 2.2. Labeling of the Phase-Contrast Image of CSCs with F-Measure Values

The standard deviations of the F-measure values were large (Figure 1d), indicating that the AI models failed to depict the correct image of the CSCs from some inputs, while they succeeded in depicting others. It was hypothesized that some phase-contrast cell images might be difficult to recognize using AI models. Thus, whether cell images with F-measure values could be classified from low to high by deep learning using a convolutional neural network (CNN) was examined (Figure 2). To obtain an AI model named FA, denoting capability of outputting F-measure value of all classes defined below, the AI was trained with 7168 pairs of Hoechst 33342-overlaid phase contrast and a pair of GFP images using pix2pix. Subsequently, another 28,672 pairs were tested using the AI model for the output images and calculation of each F-measure value (Figure 2a). The sets were then classified into 10 classes by F-measure values and used as a new dataset for CNN classification.

### 2.3. Deep Learning of Phase-Contrast Images for the Classification of CSC Images Labeled with F-Measure Values

Using the datasets, 12,249 phase-contrast images were subjected to training to create 12 AI models of classification. AI models 1–6 were tested for accuracy of classification using 128 types of phase-contrast images (Figure 2b). Each F-measure class from 0–9 included 18, 19, 26, 15, 21, 17, 6, 5, 1, and 0 images, respectively. AI model 1, which was developed using Visual Geometry Group-16 (VGG16) without transfer learning, showed an accurate classification rate of 0.4. This was very low; however, the rate increased in AI models 3 and 5, which allowed the acceptance of the number of true classes by 1 and 2. There was also an increase using AI models 2, 4, and 6, which were constructed using VGG16 with transfer learning, although the classification accuracy was low compared to that of AI models constructed without VGG16 transfer learning. These results indicated that the AI models were not efficient in performing classification. However, it was suggested that each class might have characteristic images that were distinguishable from the other classes. Images in the neighbor class defined by numbers might not be independent but rather relative to the image. Similar results were observed for AI models 7–12, which were trained with 26,672 types of Hoechst 33342-overlaid phase-contrast images and tested for accuracy using 2000 types of images (Figure 2c).

Next, classification accuracy was analyzed using AI models for each image class. Figure 2d shows the AI models trained using the phase contrast overlaid with Hoechst 33342. Accuracy showed a diverse range among classes using AI model 7. When acceptance of the number of true classes was allowed, the range decreased. This suggested that the classification was effective for all classes.

Because the classification AI models showed the capability of classifying CSCs using phase-contrast images, the referred region was visualized by AI models during classification (Figure 3a). For example, an input class 7 phase-contrast image (Figure 3a, 2nd column) included non-CSCs (upper region) and CSCs (middle region), as shown by GFP. The output class based on the CNN classification was class 5. CSCs were referred to by both gradient-weighted class activation mapping (Grad-CAM) and Grad-CAM++, although there was a difference in the distribution and color between the analyses. An AI model referred to CSCs as well as non-CSCs in an image. AI models also referred to regions without cells in some images. Then, whether the CNN classification referred to the CSCs was determined. Overall, 128 types of input images were classified with recognition of CSCs for most classes by CNN classification AI model 1, suggesting that the CSC region in the phase-contrast image was an important object for classification (Figure 3b,c).

### 2.4. Effect of Training Pairs of GFP and Phase-Contrast Images with High F-Measure Values on Constructing an AI Model of Image-to-Image Translation

It is possible that some CSC images might be recognized easily by CGAN deep learning, as shown by the high F-measure values after testing the AI models. Therefore, it was determined that it would be interesting to analyze the performance of an AI model trained using pairs with a high F-measure value. As a result, in this study, deep learning was performed on image pairs that were selected with F-measure values for image-to-image translation. Among 28,672 datasets, as described previously, 200 sets with F-measure values of 0–0.1 were randomly chosen and placed into class 0 as the dataset (Figure 4a). Accordingly, other datasets for classes 1–9 were prepared. Then, AI was trained with the datasets of classes 5–9 for each CGAN AI model F5–9 using pix2pix, and an AI model was tested with the remaining datasets from other classes that were not used for training the AI model. For example, the AI model F5 was tested using datasets from classes 0–4 and 6–7 to obtain outputs. Figure 4b shows examples of outputs for inputs with different dataset classes. Interestingly, an increase in the F-measure value was observed when the input was tested using AI models trained with higher F-measure datasets. While an input from class 0 was used to output an image with an F-measure value of 0.01 by AI model FA, which was built using datasets from all classes, AI model F5–9 outputted images with different values (Figure 4b, first line). The AI model FA failed to depict CSCs observed in the target around an arrow. The CSCs in the target were better depicted by AI model F5–9. The depiction improved the F-measure value by approximately 10-fold.

Next, the AI models were evaluated based on recall, precision, and F-measure values. It was found that AI model F9, built using dataset class 9, showed significantly higher values of recall than AI model FA for dataset classes 0–7 (Figure 5a). The values for dataset classes 1–3 increased more than twofold. In contrast, the increase in the value of precision was limited to dataset classes 0–2 (Figure 5b). The precision value decreased slightly in the dataset classes 6–8. Thus, a significant increase in the F-measure values for the dataset classes 0–6 was observed (Figure 5d). The lower the number of dataset classes, the higher the rate of increase in the value. In dataset class 0, the value of the AI model FA was increased by about 5.7-fold by AI model F9. Interestingly, AI model F5, which was built using dataset class 5, significantly improved F-measure values for the dataset class below 5, but it failed to output images with a high F-measure value for dataset classes over 5, similar to the AI model FA. It was also found that AI models F6–8 struggled to output images with F-measure values higher than those in the dataset for each training (Appendix A).

## 3. Discussion

Image recognition technology using AI has potential to identify CSCs in phase-contrast images of cells and tissues, although the process is largely unknown. Here, the effect of differences of phase-contrast images on the output was investigated to depict CSCs using CGAN image translation. To train the AI model, the addition of nuclear images to the phase contrast using Hoechst 33342 did not increase the accuracy of depiction of CSCs. CSC images produced by the AI models included a diverse range of accuracy values to the target true images, indicating that the AI models depicted CSCs reasonably accurately from some inputs but not others. To prove the possibility that AI might uniquely recognize phase contrast with different F-measure values, phase-contrast images of CSCs were shown to be possible for CNN classification according to accuracy values of the F-measure, which was obtained by testing a large dataset using CGAN image translation. AI models trained using a high F-measure dataset for CGAN image translation produced CSC images more accurately to the target for an input with a lower F-measure value. Therefore, there could be various CSC images of phase contrast that could be efficiently recognized by the AI model. Compared to the non-selective dataset for training, the use of high F-measure values as the training dataset was effective to obtain accurate AI models depicting CSC images from phase contrast.

In deep learning of CSC morphology, phase-contrast images in the training dataset are important for the quality of the AI model. For live CSC morphology to be mapped by CGAN image translation, CSC images from several different culture conditions are efficient [14]. For CSCs in tumor tissue, phase-contrast images labeled with nuclei using Hoechst 33342 are slightly efficient [15]. Accordingly, here, cell images from variable conditions and labeled cell nuclei were used for images with Hoechst 33342-overlaid phase contrast. In contrast, the efficiency did not increase in live CSC cultures. Previous reports have suggested that color information in images affects CNN classification [16,17]. In these previous studies, colored objects were used in the original images, whereas cultured CSCs had no color and exhibited transparency with tough morphology for a trainee to distinguish CSCs from non-CSCs. Because CSCs have scant cytoplasm [7], the addition of nuclear information with a color to the phase-contrast image did not allow visual differentiation. Therefore, Hoechst 33342-overlaid phase contrast did not increase the accuracy.

Although deep learning was performed using variable CSC images with the expectation of creating a high-accuracy AI model, CGAN image translation depicted CSC images with low-to-high accuracy using a wide range of F-measure values. The AI model might have preferred some phase-contrast images but not others for depicting CSC images. This possibility was supported by the demonstration of CNN classification for phase-contrast images of CSCs. The CNN classification of images into 10 classes by F-measure values was not sufficiently accurate; however, most of the test phase-contrast images were classified within the next class. Interestingly, the AI model referred to CSCs among objects, such as culture dishes, non-CSCs, and CSCs in an image. Non-CSCs are often referred to in CNN classification of images with lower F-measure values. The AI model may use non-CSCs for the classification of these classes. These results suggested that there was a difference in the phase-contrast images of cultured CSCs that was recognized by AI models. What if this information was utilized for the selection of a deep learning dataset for CGAN image translation? The present attempt to use selected images with high F-measure values to train AI created a novel AI model that significantly improved the accuracy, especially in images with lower F-measure values. Interestingly, the accuracy of depiction did not extend to a higher F-measure value than that used for selecting images for deep learning. This implied that a CSC image was recognized differently by the AI models of CGAN image translation. It could be possible to achieve an AI model trained using the selected dataset based on the F-measure values, which were obtained through a large dataset tested by an AI model trained using another unselected dataset.

Thus, the application of AI in biological imaging has emerged. Most studies have aimed to develop, modify, combine, and apply algorithms of neural networks for an AI model with high accuracy [18,19,20]. For example, U-Net was developed for cells [21]. A spheroid cell cluster was segmented using a CNN and generative adversarial networks [22]. The importance of the dataset for training was emphasized for the desired AI model, which was not affected by simple changes in the images, such as rotated images. It is possible that including a selected dataset suitable for networks in CGAN translation is effective for deep learning. However, it is difficult to determine which dataset is most effective. Here, differences between phase-contrast images of CSCs were not observed, although differences were clear in F-measure values among phase-contrast images. AI model F9 could bring a kind of expertise in the recognition of morphologic character of CSCs in terms of accuracy which cannot be obtained by other AI models trained simply using large datasets. An AI model built through the workflow of F9 is useful for supporting cancer diagnosis, alerting to the presence of CSCs in a specimen which is expected to be developed. Success of AI model F9 also suggested the presence of undefined structure which was referred to by F9 for CSCs. CSCs might maintain the structure while changing the cell morphology. So far, the efficiency of the workflow of deep learning for CSC morphology has been demonstrated. It would be interesting to apply this methodology to other cell types, such as iPS and ES cells. Further studies are required to determine how AI models can recognize CSC images.

## 4. Materials and Methods

### 4.1. Cell Culture

The miPS-LLCcm cell line, a mouse CSC model, was derived from iPS-MEF-Ng-20D-17 [6]. The cells were cultured in a medium (Dulbecco’s modified Eagle’s medium high glucose:LLC-cell-culture-conditioned medium, 1:1; 1 × non-essential amino acids, 10% fetal bovine serum, 1% penicillin/streptomycin, and 100 μM 2-mercaptoethanol) at 37 °C in a 5% CO_2_ incubator. The reagents for cell culture were purchased from the FUJIFILM Wako Pure Chemical Corporation (Osaka, Japan). Culture dishes were pre-coated with 0.1% porcine skin gelatin (Millipore Sigma, St. Louis, MO, USA) as described previously [7]. 

### 4.2. Microscopy

Cells were incubated with medium containing 4 μg/mL Hoechst 33342 for 10 min. After changing the medium to without Hoechst 33342, cell images were obtained using the fluorescence microscope BZ-X800 (KEYENCE, Osaka, Japan). The GFP image was visualized using a filter (excitation: 450–490 nm, emission: 500–550 nm) with an exposure time of 1.5 s in channel CH1. A phase-contrast image was obtained with an exposure time of 1/60 s in channel CH2. The Hoechst 33342 fluorescence image was visualized using a filter (excitation: 320–360 nm, emission: 435–485 nm) with an exposure time of 1/12 s in channel CH4. All images were acquired at a resolution of 1920 × 1440 px and were saved as TIFF files. The CH2 image overlaid with the CH4 image was obtained using the BZ-800 analyzer v.1.1.2.4 (KEYENCE).

### 4.3. Image Processing and AI

Machine learning was performed on an Ubuntu OS (Canonical Ltd., London, UK) with GPU-accelerated software environments (NVIDIA Corp., Santa Clara, CA, USA) built on a personal computer with a GPU. For image-to-image translation with CGAN, TensorFlow implementation of the pix2pix code was obtained from https://github.com/affinelayer/pix2pix-tensorflow on 17 May 2019. A pair of CH1 and CH2 images was processed to create 35 new pairs of image files with a resolution of 224 × 224 px. These pairs were used as inputs for pix2pix. To evaluate the AI output, the precision (*P*), recall (*R*), and F-measure values were calculated. In brief, an identical region of AI output and the target images was employed for *P* and *R*. The F-measure is defined by Equation (1):(1)$$F-\mathrm{measure}=\frac{2\mathrm{PR}}{P+R}$$

For image classification and visualization, the PyTorch v1.7.1 (https://pytorch.org/) machine learning framework was used, which supports the CNNs of VGG16 [23] and residual neural network 50 [24] for classification. Grad-CAM [25] and Grad-CAM++ [26], which are methods for visualizing active regions during classification by a CNN, were used with the PyTorch library.

### 4.4. Statistical Analysis

Student’s *t*-test was used to determine the statistical significance of the differences between two groups.

## 5. Conclusions

The deep learning ability of CSCs in culture was investigated. The CGAN AI model identified CSCs accurately for some phase-contrast images, but inaccurately for others. The CNN classification AI model suggested the presence of various phase-contrast images for CSCs. The accuracy of the CGAN AI model was affected by the training dataset with a high F-measure value obtained using the previous CGAN AI model. The deep learning workflow could be useful for improving the quality of segmentation by CGAN.

## Figures and Tables

**Figure 1 ijms-24-05323-f001:**
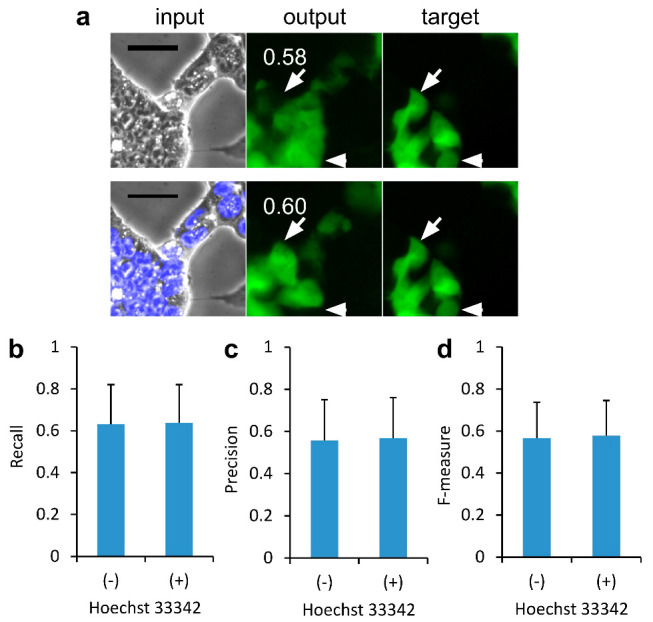
Comparison of the output of cancer stem cell (CSC) images between artificial intelligence (AI) models trained in the presence and absence of cell nucleus images. (**a**) Output example of AI models. The test input image of phase contrast with or without Hoechst 33342 was subjected to depiction as a fluorescence image. Scale bars = 100 μm. The number in the output represents the F-measure value. The output image was compared with each target: (**b**) recall, (**c**) precision, and (**d**) F-measure. Mean ± S.D., *n* = 1000.

**Figure 2 ijms-24-05323-f002:**
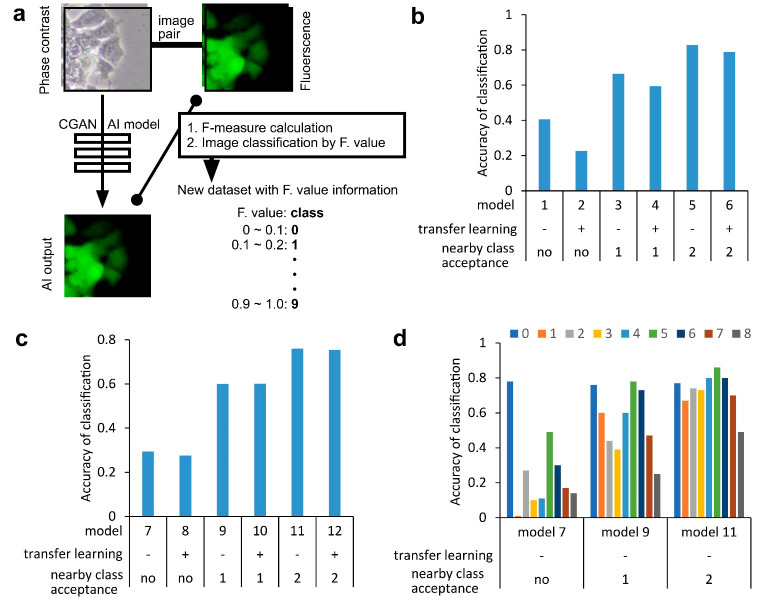
Classification of the CSC images. (**a**) A scheme of CSC image classification using F-measure values obtained through the comparison of output images of a conditional generative adversarial network (CGAN) AI model and the target. A new dataset with image classes was constructed by using the F-measure information. The accuracy of classification into 10 classes by AI models trained using phase-contrast images of mouse-induced pluripotent stem-Lewis lung cancer cm without (**b**) and with (**c**) Hoechst 33342-overlaid images was determined. Deep learning was performed using Visual Geometry Group-16. The accuracy was calculated by counting the number of outputs correctly classified into the true class among total test inputs; (**d**) The detailed classes of outputs by AI models shown in (**c**). The column number represents the class.

**Figure 3 ijms-24-05323-f003:**
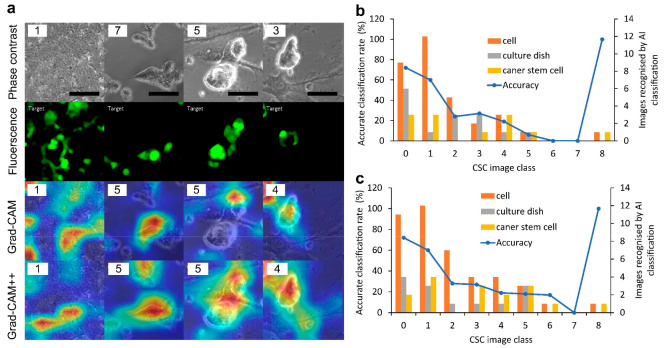
Visualization of convolutional neural network (CNN) classification. (**a**) Output example of an image. Input images of phase contrast belong to the class as labeled on the top left of each image. For example, class 7 is labeled as 7. Fluorescence images paired with phase contrast show the CSC region. The output class by CNN classification using gradient-weighted class activation mapping (Grad-CAM) and Grad-CAM++ is labeled on the top left of each image. The region of the heat map color indicates the position where the CNN shows the referring objects during deep learning. A darker shade of red indicates that the probability of the position is higher. Scale bars = 100 μm; (**b**) Objects recognized by CNN classification using Grad-CAM. Objects of cells, culture dishes, and CSCs, which are marked with red, were counted for each image; (**c**) Objects recognized by CNN classification using Grad-CAM++.

**Figure 4 ijms-24-05323-f004:**
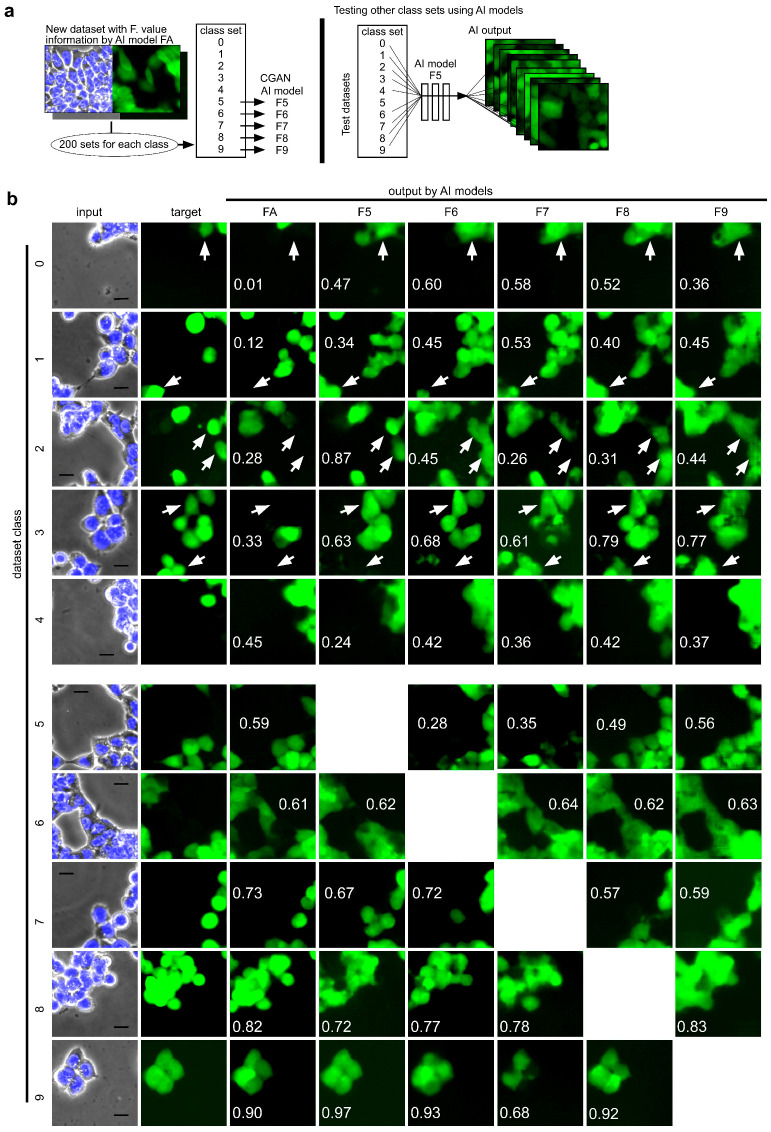
Comparison of output CSC images depicted by AI models trained using images with the same class of F-measure values. (**a**) A scheme of dataset preparation for training and testing the AI model. The new dataset included 28,672 pairs with F-measure values that were obtained using the AI model trained with 7168 pairs (AI model FA). The class set included 200 pairs that were selected by F-measure value. AI models F5–6 were trained with F-measures specifically; (**b**) CSC image depicted from a Hoechst 33342-overlaid phase-contrast image. The input with various F-measure values was tested for the output using various AI models. Arrows were placed for visual assistance of comparison between outputs. Numbers in the output represent the F-measure value calculated using target and output images. Scale bars = 50 μm.

**Figure 5 ijms-24-05323-f005:**
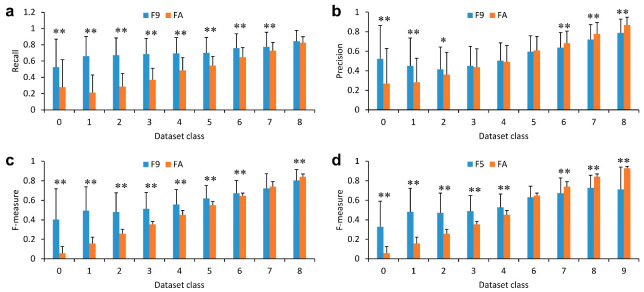
Evaluation of the AI model trained using images with the same class of F-measure values. The output image by AI model F9 was compared with the target by calculating the values of (**a**) recall, (**b**) precision, and (**c**) F-measure; (**d**) F-measure values of output images by AI model F5. Mean ± S.D., *n* = 200, ** *p* < 0.01, * *p* < 0.05.

## Data Availability

Image datasets used in this study are available on Google Drive (https://drive.google.com/drive/folders/1UgzhHkDnQ6aw-TALcrtVcX6PvawWUnV0?usp=sharing, accessed on 26 February 2023).

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
