# Peer review of "Deep Learning of Phase-Contrast Images of Cancer Stem Cells Using a Selected Dataset of High Accuracy Value Using Conditional Generative Adversarial Networks"

_ijms, 2023, doi:10.3390/ijms24065323_

Round 1

Reviewer 1 Report

In this study, the authors evaluated Deep learning of phase-contrast images of cancer stem cells using a selected dataset of high accuracy value using conditional generative adversarial networks.

Generally, all research activities were performed in detail and in the proper manner, so the reviewer had no suggestions.

The abstract is an accurate summary of the research and results.

The introduction is concise with all necessary data accompanied by adequate references.

The material and methods are completely explained and very well written.

The research questions are valid, the sample size is sufficient, and the methods and study design are appropriate for answering the research question.

The experiments do have appropriate controls. Besides, the reporting of the methods, including any equipment and materials, is sufficiently detailed that the research might be reproduced. All statistical tests were used appropriately and correctly were reported.

The results correspond to the objectives of the study. This section is excellently written with all the necessary and concise accompanying explanations. The results do support the conclusions.

The figures are clear and they accurately represent the results. The figures are in a satisfactory resolution so that all the mentioned and explained details are visible.

The discussion part is complete and very well written.

The references used are carefully selected and also up-to-date. There aren't inappropriate citations, as well as too many citations to the author's own articles.

The language is clear and understandable.

Reviewer 2 Report

Thank you for the opportunity to review this manuscript.

In this study, the authors discuss the role of artificial intelligence (AI) technology in the identification of cancer stem cells in culture and tissue specimens.

In my opinion, this study may potentially offer novel criteria to guide cancer research in the future.

However, I recommend that the authors present a detailed discussion regarding the future application of standards such as F9 created by AI technology.
